# Prediction models for the prediction of unplanned hospital admissions in community-dwelling older adults: A systematic review

Jet H. Klunder[1,2]*, Sofie L. Panneman[1], Emma Wallace[3], Ralph de Vries[4], Karlijn J. Joling[2,5], Otto R. Maarsingh[1,2], Hein P. J. van Hout[1,2,5]

1 Department of General Practice, Vrije Universiteit Amsterdam, Amsterdam UMC, Amsterdam, the Netherlands, 2 Amsterdam Public Health, Aging & Later Life, Amsterdam, the Netherlands, 3 Department of General Practice, University College Cork, Cork, Ireland, 4 Medical Library, Vrije Universiteit, Amsterdam, the Netherlands, 5 Department of Medicine for Older People, Vrije Universiteit Amsterdam, Amsterdam UMC, Amsterdam, the Netherlands

* j.h.klunder@amsterdamumc.nl

**Data Availability Statement:** All relevant data are within the paper and its Supporting Information files.

## Abstract

### Background

Identification of community-dwelling older adults at risk of unplanned hospitalizations is of importance to facilitate preventive interventions. Our objective was to review and appraise the methodological quality and predictive performance of prediction models for predicting unplanned hospitalizations in community-dwelling older adults

### Methods and findings

We searched MEDLINE, EMBASE and CINAHL from August 2013 to January 2021. Additionally, we checked references of the identified articles for the inclusion of relevant publications and added studies from two previous reviews that fulfilled the eligibility criteria. We included prospective and retrospective studies with any follow-up period that recruited adults aged 65 and over and developed a prediction model predicting unplanned hospitalizations. We included models with at least one (internal or external) validation cohort. The models had to be intended to be used in a primary care setting. Two authors independently assessed studies for inclusion and undertook data extraction following recommendations of the CHARMS checklist, while quality assessment was performed using the PROBAST tool. A total of 19 studies met the inclusion criteria. Prediction horizon ranged from 4.5 months to 4 years. Most frequently included variables were specific medical diagnoses (n = 11), previous hospital admission (n = 11), age (n = 11), and sex or gender (n = 8). Predictive performance in terms of area under the curve ranged from 0.61 to 0.78. Models developed to predict potentially preventable hospitalizations tended to have better predictive performance than models predicting hospitalizations in general. Overall, risk of bias was high, predominantly in the analysis domain.

**Funding:** The authors received no specific funding for this work.

**Competing interests:** The authors have declared that no competing interests exist.

## Conclusions

Models developed to predict preventable hospitalizations tended to have better predictive performance than models to predict all-cause hospitalizations. There is however substantial room for improvement on the reporting and analysis of studies. We recommend better adherence to the TRIPOD guidelines.

## Background

In the Netherlands, approximately one in five older adults is admitted to hospital each year [1]. Moreover, hospital admission rates in ED patients aged 65 years and older are twice as high as those in ED patients aged <65 years [2]. When hospitalized, older adults are at high risk of experiencing adverse events such as hospital-associated infections and delirium, causing lengthy hospital stays [3, 4]. In addition, hospitalizations pose a significant risk to the functional ability of older adults, whereas 30% of older patients experiences loss of independence in activities of daily living (ADL) after hospital admission [5].

Older adults account for a large proportion of hospitalized adults, which is likely to increase with the aging population, causing overcrowding of emergency departments (EDs) and hospital wards [6, 7]. Overcrowded EDs have been described as a global health problem having negative effects on patients (e.g. treatment delay), healthcare staff (e.g. stress) and the healthcare system (e.g. increased length of stay in ED as well as in hospital wards) [8]. Taking into account that a large proportion of hospitalizations and ED visits in older adults is considered preventable [9], it seems crucial to timely identify older adults at risk of hospitalization to assess possible preventive measures. This would not only increase patient's health and quality of life, but also relieve pressure on secondary and tertiary care, resulting in a decrease in overall health care costs [10].

Prediction models can be used to identify community-dwelling older adults at risk for unplanned hospital admissions. By defining and combining important predictors of future emergency care use, preventive interventions can be targeted at high risk individuals [11]. Several prediction models for the prediction of unplanned hospitalizations have been developed and two systematic reviews on this topic have previously been published. However, these reviews included studies in adults of all ages or only included easy to apply case-finding instruments [12, 13]. Furthermore, these reviews were published over seven years ago. In an era of personalized and precision medicine, interest in and the number of prediction models have grown rapidly [14, 15]. Moreover, with the emergence of big data, attention has grown towards different modelling techniques beside traditional regression methods, such as machine learning (ML). Despite guidelines as the Transparent Reporting of a multivariable prediction model for Individual Prognosis Or Diagnosis (TRIPOD) [16], quality of methodology and reporting of clinical prediction model studies is however often insufficient [17, 18].

We carried out a systematic review of validated prediction models for predicting unplanned hospital admissions in community-dwelling older adults (≥65 years). Our objective was to describe characteristics of the models' development, the predictors included in the final models, the predictive performance, and to appraise methodological quality of the included models.

## Methods

This review is reported according to the Preferred Reported Items for Systematic Reviews and Meta-Analyses (PRISMA) Statement [19]. The study protocol has been registered on the

International Prospective Register of Systematic Reviews (PROSPERO, registration number: CRD42020207305).

## Search strategy, study selection and data-extraction

We conducted systematic searches in the bibliographic databases PubMed, Embase.com and CINAHL (Ebsco) in January 2021, in collaboration with a medical information specialist. The following terms were used (including synonyms and closely related words) as index terms or free-text words: "Hospital admission", "Patient admission", "Unplanned", "Aged", "Older adults", "Prediction". We applied a validated search filter for finding clinical prediction model studies [20]. The full search strategies are provided in S1 File.

As previously mentioned, two systematic reviews on this topic have been published. Wallace et al. carried out a systematic literature search in February 2014 on risk prediction models to predict emergency admissions in community-dwelling adults [13]. O'Caoimh et al. reviewed short case-finding instruments, published up and until November 2014, for community-dwelling older adults (> 50 years) at risk for multiple adverse outcomes, of which hospitalization was one [12]. To provide a complete overview of available prediction models our search was restricted to August 2013 through January 2021 and we added the models described in the previous reviews that fulfilled the eligibility criteria of this systematic review.

The references of the identified articles were searched for relevant publications. Duplicate articles were excluded.

Studies were included if they met the following criteria:

i. Population: community-dwelling older adults, aged 65 years and over

ii. Intervention: prognostic prediction models derived from retrospective or prospective cohort studies and containing at least one validation cohort

iii. Comparator: not applicable

iv. Outcome: one or more unplanned hospitalizations (defined as unplanned overnight stay in hospital). Studies that had admission to the ED as part of their outcome of interest (i.e. combined endpoints) were also included

v. Timing: admission to hospital within any time period

vi. Setting: prediction models intended to be used in primary care

We excluded studies if the prediction models:

i. were contingent on an index hospital admission or ED visit (i.e. readmission models)

ii. studied hospitalizations for specific conditions (e.g. falls or congestive heart failure) as primary outcome

iii. were intended to be used in the ED

iv. were developed in specific populations (e.g. patients in palliative care or with psychiatric conditions), with the exception of participants with sensory impairments, because of high prevalence in the older population [21]

Studies that assessed risk factors only and did not build a prediction model, studies that were not developed to specifically predict unplanned hospitalizations, such as models that identify frailty, and studies published in languages other than English, Dutch, German, French, Italian and Spanish were also excluded.

All records were deduplicated in Endnote v9.1, and consequently exported to the Rayyan web app for title and abstract screening and study selection [22]. After study selection, data extraction was performed using a standardized form following the recommendations of the Checklist for critical Appraisal and data extraction for systematic Reviews of prediction Modelling Studies (CHARMS; S2 File) [23]. Both selection and data extraction phases were independently conducted by two reviewers (JK and SP). Any disagreements were resolved through a consensus procedure or by third review (OM, KJ, HvH). Additional data were sought from authors, when necessary.

Due to heterogeneity of the prediction models, meta-analysis was not possible. We therefore narratively summarized each unique prediction model on study population, predictors, number of outcomes and predictive performance. For clarity reasons, regression models and machine learning models were presented separately. Predictive performance was assessed as model discrimination using the area under the ROC curve (AUC) with 95% confidence intervals. Higher AUC values indicate better discriminatory ability. An AUC of 0.7–0.8 reflects fair discrimination, whereas a model with AUC $\geq$ 0.8 represents good discrimination [24].

## Methodological quality assessment

The Prediction model Risk of Bias ASsessment Tool (PROBAST; S2 File) was used to assess risk of bias and applicability, of which the latter addresses whether the primary study matches the review question [25]. PROBAST rates study methodology and applicability to the review question as being at "high", "low" or "unclear" risk of bias based on a predetermined set of questions and scoring guide [26].

In addition, we calculated the number of events per variable (EPV) for each model. The number of EPV is the number of outcome events divided by the number of candidate predictors assessed in the multivariable modelling [27]. Studies with an EPV <10 are generally subject to overfitting, therefore an EPV of >20 is recommended. Prediction models developed using ML techniques often require higher EPVs (often >200) to minimize overfitting [26].

## Results

### Study selection

The literature searches yielded a total of 16,098 citations (Fig 1). After removing duplicates 8,820 references remained. Additionally, twenty-three articles were identified by checking the reference lists of relevant studies. Full texts were retrieved for 170 studies of which ten met all inclusion criteria. Additionally, a total of nine studies were included from the previously published systematic reviews (Tables 1 and 2).

### Description of included studies

Of the 19 studies included, the majority were developed in the United States (n = 10) [28–37] and two in Italy [38, 39]. The other studies were developed in the United Kingdom [40], Ireland [41], Canada [42], Sweden [43], Spain [44], Taiwan [45], and South-Korea [46]. Twelve studies included participants aged $\geq$65 years [29, 33, 35–39, 41, 42, 44–46], the remaining studies used a higher age as inclusion criterion with 81 years [30] as the highest minimum age for inclusion. Total sample sizes ranged from 150 [34] to 1,095,613 [39] participants. Two studies were developed in patients receiving home or community care [34, 41], and one study developed a prediction model in older adults with a vision and/or hearing impairment [29].

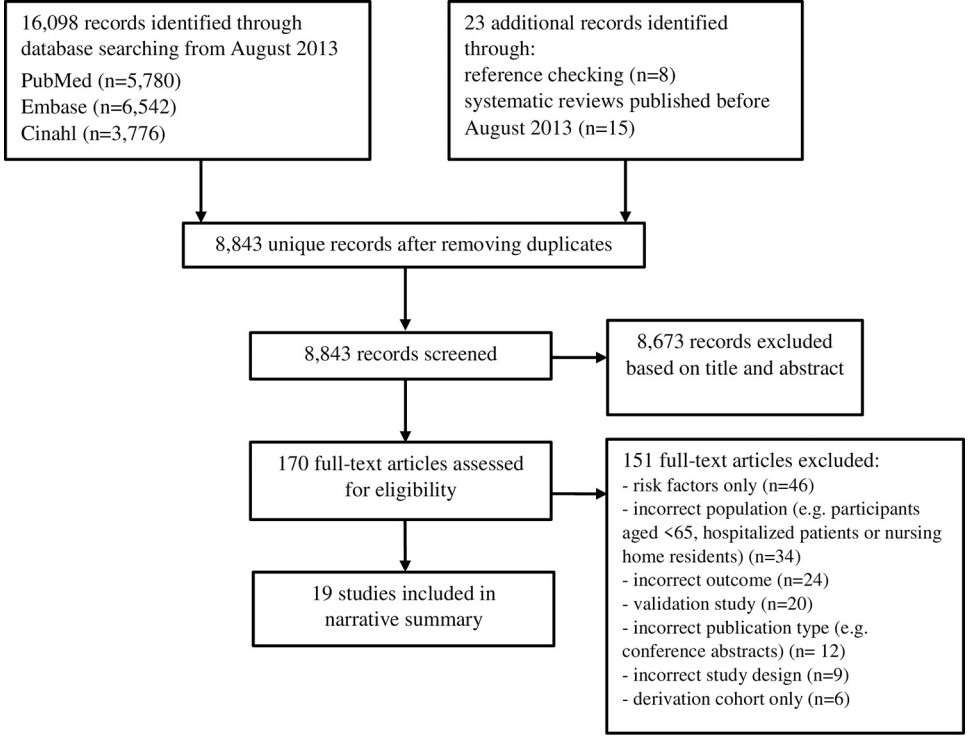

**Fig 1. PRISMA flow diagram of included risk prediction models.**

Eight studies developed their model using administrative or electronic medical record data [31, 32, 34, 39, 43–46]. Eight studies used survey data to develop their model [28–30, 33, 36, 38, 40, 41], and three models were developed using both [35, 37, 42].

Various outcomes were assessed in the development of the prediction models. Two studies validated their models for more than one outcome (i.e. unplanned hospitalizations and potentially preventable hospitalizations, separately) [37, 39]. Two models predicted a combined endpoint of any hospitalization or ED visit [34, 36]. Fourteen studies assessed unplanned hospitalizations as single endpoint [29–33, 37–45], two studies predicted multiple hospitalizations within a specific time period [28, 35], and three studies presented a model for potentially preventable hospitalizations [37, 39, 46]. Two out of these three studies defined admissions as potentially preventable based on the principal diagnosis on admission [37, 46]. The third study did not report its definition for preventable admissions [39]. The prediction horizon ranged from 4.5 months [30] to 4 years [28]. The majority of studies (n = 12) were developed to predict the outcome within 12 months [29, 31, 32, 36, 37, 39–41, 43–46].

## Variables used in prediction models

The number of predictors included in the final model ranged from 3 [36] to 38 [43]. The variables most frequently included in the final models were previous hospital admission (n = 11) [28, 29, 31, 35, 36, 38, 40, 42–44, 46], age (n = 11) [28, 31–33, 37–39, 43–46] and sex or gender (n = 8) [28, 32, 33, 35, 37, 38, 43, 44] (Table 3). Twelve studies included one or more specific diseases in the final model, of which cardiovascular diseases (e.g. coronary artery disease, heart failure, or hypertension) were most frequently included (n = 11) [28–30, 33, 36, 39, 40, 42–45]. The most frequently included cardiovascular predictor was ischemic heart disease (n = 7) [28, 29, 33, 39, 42–44]. Diabetes was included in seven models [28, 30, 33, 35, 42–45]. Other

**Table 1. Prediction models developed using regression methods.**

| First author + reference | Acronym | Modelling method | Population + Setting | Derivation, n | Validation, n | Data used for final model | Outcome | Number of outcome events, n (%) | AUC (95% CI) | Predictors in final model |
|---|---|---|---|---|---|---|---|---|---|---|
| Boult [28] | Pra (Probability of repeated admissions) | Logistic regression | Non-institutionalized patients aged ≥70, United States, 1984–1990 | 2942 | 2934 (split sample) | Predictors = Self-report data from longitudinal study of aging Outcome = Medicare program records | ≥2 hospital admissions in 4 years | Internal validation = 669 (22.7%) | Internal validation = 0.61 | 1. Age 2. Sex 3. Self-rated health 4. Availability of informal caregiver 5. Diagnosis of coronary artery disease 6. Diagnosis of diabetes 7. Hospital admission in previous year 8. ≥6 doctor visits in previous year |
| Deardorff [29] | | LASSO Logistic regression | Community dwelling Medicare beneficiaries with hearing and/or vision impairment aged ≥65, United States, 1999–2006 | 15,999 | N/A bootstrap validation in full cohort | Predictors = Medicare Current Beneficiary Survey data Outcome = Claims data | Hospital admission in 12 months | Derivation = 2567 (16.0%) Validation = N/A | Derivation = 0.72 Validation = 0.72 | 1. Number of inpatient admissions in previous year 2. Number of ED visits in previous year 3. ADL difficulty level 4. Poor self-rated health 5. History of myocardial infarction 6. History of stroke 7. History of non-skin cancer |
| Freedman [30] | | Logistic regression | Patients with a Kaiser Permanente health plan aged ≥81, United States, 1993 | 1873 | 1872 (split sample) | Predictors = Self-administered questionnaire Outcome = Health plans computerized records system | Hospital admission in 4.5 months | Derivation = NR Validation = NR | Derivation = 0.69 Validation = 0.63 | 1. Heart trouble 2. Limited physical independence 3. Interaction of 1. and 2. 4. Need help preparing meals 5. Diabetes |
| Inouye [31] | | Logistic regression | Patients aged ≥70 in 2 primary care clinics, United States, 2003–2006 | 1932 | 1987 (split sample) | Predictors and outcome = Administrative data | Unplanned hospital admission within 12 months | Derivation = 299 (15%) Validation = 328 (17%) | Derivation = 0.72 Validation = 0.73 | 1. CCI ≥2[1] 2. Hospitalization in previous year 3. Primary care visits ≥6 in previous year 4. Age ≥85 5. Unmarried |
| Kan [32][2] | | Full model approach | Patients enrolled in a local Medicare Advantage Health Maintenance Organization plan, United States, 2011–2013 | 16,705 | NR Temporal validation | Three models 1. Based on claims data 2. Based on EHR-structured data 3. Based on EHR-structured and EHR-unstructured data | ≥1 hospitalization within 12 months | Predictive model = 3174 (19.0%) | Predictive model = Model 1 = 0.70 Model 2 = 0.70 Model 3 = 0.71 | 1. Age 2. Sex 3. Race 4. Number of major ADGs[2a] 5. Number of hospital dominant conditions[2b] 6. Number of frailty risk factors[2c] |

(*Continued*)

**Table 1.** (Continued)

| First author + reference | Acronym | Modelling method | Population + Setting | Derivation, n | Validation, n | Data used for final model | Outcome | Number of outcome events, n (%) | AUC (95% CI) | Predictors in final model |
|---|---|---|---|---|---|---|---|---|---|---|
| Kim [46] | | Logistic regression | Insured adults aged ≥ 65, South Korea, 2011–2012 | Total sample: 113,612 | NR (split sample + bootstrap) | Predictors and outcome = Routinely collected claims data | Potentially avoidable hospitalization within 12 months | Total sample = 2856 (2.5%) | Derivation = 0.77 (0.76–0.79) Validation = 0.78 (0.77–0.80) | 1. Age 2. Living area 3. Insurance 4. No. chronic conditions 5. Polypharmacy 6. Disability 7. Hospitalization in past year 8. Total health expenditures in past year |
| Kurichi [33] | | Logistic regression | Medicare Beneficiaries aged ≥65, United States, 2001–2007 | 15,606 | 7801 (split sample) | Predictors = routinely collected survey data Outcome = claims files Two models due to collinearity: 1. ADL limitation 2. IADL limitation | Hospital admission within 3 years | NR | Development = ADL limitation: 0.67 iADL limitation: 0.67 Validation = ADL limitation: 0.67 iADL limitation: 0.67 | 22/23 variables:[3] 1. Sociodemographics (n = 6) 2. Self-reported health conditions (n = 13) 3. Vision impairment 4. Smoking 5. (I)ADL stage 6. Proxy responded (IADL model only) |
| Lin [45] | | Logistic regression | Subjects aged ≥65 with at least 1 outpatient visit in 2008, Taiwan, 2008–2009 | 133,726 | 44,560 (split sample) | Predictors and outcome = claims files from national health insurance institute | Hospital admission within 1 year | Derivation = 25,541 (19.1%) Validation = 8511 (19.1%) | Development = 0.64 (0.64–0.65) Validation = 0.64 (0.63–0.65) | 1. Age 2. Education 3. COPD 4. Heart disease 5. Diabetes 6. Cancer (with or without metastases) 7. Chronic kidney disease 8. ED visit in past year 9. Received home care in past year |
| López-Aguilà [44] | | Logistic regression | Patients in primary care aged ≥65, Spain, 2006–2009 | 28,430 | NR | Predictors = Clinical records of primary care centers, pharmacy database, and hospital discharge records Outcome = Hospital discharge records | Unplanned hospital admission in 12 months | Derivation = 2103 (7.3%) Validation = NR | Derivation = 0.78 Validation = 0.76 | 1. Sex 2. Age 3. COPD 4. Heart failure 5. 5 or more concurrent diagnoses 6. 4 or more prescribed drugs 7. 2 or more emergency admissions[4] 8. 2 or more planned admissions[4] 9. 9 or more days of cumulative stay[4] |

(*Continued*)

**Table 1.** (Continued)

| First author + reference | Acronym | Modelling method | Population + Setting | Derivation, n | Validation, n | Data used for final model | Outcome | Number of outcome events, n (%) | AUC (95% CI) | Predictors in final model |
|---|---|---|---|---|---|---|---|---|---|---|
| Lyon [40] | EARLI (Emergency Admission Risk Likelihood Index) | Logistic regression | Patients in general practices aged ≥75, England, 2002–2003 | 3032 | 500 (split sample + bootstrap) | Predictors = Questionnaire Outcome = Hospital Episodes Statistics data | Unplanned hospital admission in 12 months | Derivation = 696 (23.0%) Validation = NR | Derivation = 0.70 (0.67–0.72) Validation = • Bootstrap validation = 0.69 • Split sample validation = 0.67 (0.63–0.71) | 1. Heart problems 2. Leg ulcers 3. Get out of the house without help 4. Problems with memory or get confused 5. Emergency hospital admission in last 12 months 6. Overall state of health |
| Marcusson [43] | | Logistic regression / LASSO | Patients in primary care aged ≥75, Sweden, 2015–2017 | 20,364 | Internal validation = 20,364 (split sample) External validation: 1) 51,104 (sample with ages 65–74) 2) 38,121 (different time period) | Predictors and outcome = computerized information system of the County Council of Östergötland. | Unplanned hospital admission within 12 months | Derivation = 4130 (20.3%) Validation = Split sample: 4037 (19.8%) External validation: NR | Derivation: NR Internal validation: 0.69 (0.68–0.70) External validation: 1) 0.68 (0.67–0.69) 2) 0.68 (0.67–0.69) | 38 predictors = 1. Sex 2. Age 3. Number of non-physician visits 4. Number of physician visits, 5. Number of previous in-ward hospital stays 6. Number of ED visits 7. Signs/symptoms and medical diagnoses (n = 32) |
| Mazzaglia [38] | | Logistic regression | Persons in primary care aged ≥65, Italy, 2003–2004 | 2470 | 2926 (external validation) | Predictors = Questionnaire answered by primary care physician, registries of the regional health system of Tuscany Outcome = Registries of the regional health system of Tuscany | Hospitalization in 15 months | Derivation = 445 (18.0%) Validation = 504 (17.2%) | Derivation = 0.68 (0.66–0.71) Validation = 0.67 (0.65–0.70) | 1. Number of positive responses to screening test[5] 2. Age 3. Sex 4. Hospitalization in previous 6 months 5. ≥5 prescriptions |
| Mishra[6] [34] | | Mixed effects logistic regression Full model approach | Residents at an Aging-in-Place facility, United States, 2011–2019 | N/A | 150 participants, 4495 individual assessments | Predictors and outcome = routinely collected assessments in EMR every 6 months | ED visit or hospital admission within 6 months | NR | 0.72 (0.65–0.79) | Geriatric assessments: 1. ADL[6a] 2. IADL[6b] 3. Depressive symptoms[6c] 4. Cognition[6d] 5. Mental health[6e] 6. Physical health[6e] |
| O'Caoimh [41] | RISC (Risk Instrument for Screening in the Community) | Iterative process of item generation and reduction using literature searches and focus groups with public health nurses (PHN) | Community-dwelling adults ≥65 under follow-up by PHN, Ireland, 2012–2013 | N/A | 801 | Predictors = PHN review and additional GP information Hospitalization = Data from hospital enquiries | Acute admission to an acute hospital within 12 months | Validation = 142 (17.7%) | Validation = 0.61 (0.55–0.66) | 1. Age 2. Gender 3. Living arrangement 4. Presence and magnitude of concern for PHN across 3 domains: • Mental state • ADL • Medical/physical state 5. Ability of caregiver to manage (according to PHN) |

(*Continued*)

**Table 1.** (Continued)

| First author + reference | Acronym | Modelling method | Population + Setting | Derivation, n | Validation, n | Data used for final model | Outcome | Number of outcome events, n (%) | AUC (95% CI) | Predictors in final model |
|---|---|---|---|---|---|---|---|---|---|---|
| Reuben [35] | | Logistic regression | Medicare beneficiaries aged ≥65, United States, 1988–1992 | 2569 | 2569 (split sample) 10-fold cross-validation | Predictors = Interviews, physical examination, and laboratory testsThree models developed = 1. Self-reported prior hospitalizations only 2. Self-report variables 3. Self-report, physical examination, and laboratory variables Outcome = claims data | High utilization (≥11 hospital days in 3 years) | Full cohort = 1243 (24.2%) | Full cohort (after cross-validation) = 1. 0.60 2. 0.68 3. 0.69 | Self-reported predictors: 1. Any hospitalization in previous year 2. Any hospitalization in year before that 3. Male gender 4. Fair or poor health 5. Not currently working 6. Little participation at religious services 7. Need help with bathing 8. Unable to walk a mile 9. Diabetes, sugar in urine, or high blood sugar, 10. Taking loop diuretics Laboratory results: 11. Serum albumin 12. Serum iron |
| Roos [42] | | Logistic regression | Insured participants aged ≥65 years, Canada 1970–1973 | 1518 | 1518 (split sample) | Predictors = Three models were compared 1. Administrative data only 2. Interview data 3. Administrative and interview data Outcome = claims data | Hospital admission within 24 months | NR | NR | Interview questions: 1. Self-rated health 2. Reported conditions of arthritis, diabetes, chest 3. Reported undergoing ≥1 treatment 4. Amount of time spent in hospital in last year Administrative data: 5. Living with spouse 6. Prior hospital utilization in last year 7. Prior ambulatory utilization in last year |

(*Continued*)

**Table 1.** (Continued)

| First author + reference | Acronym | Modelling method | Derivation, n | Validation, n | Population + Setting | Data used for final model | Outcome | Number of outcome events, n (%) | AUC (95% CI) | Predictors in final model |
|---|---|---|---|---|---|---|---|---|---|---|
| Shelton [36] | CARS (Community Assessment Risk Screen) | Logistic regression | 411 | 1054 (external validation) | Medicare patients with ≥1 specified characteristic and ≥65 years, United States, 1993–1995 | Predictors = telephone interviews, mailed questionnaires; Outcome = claims files (hospitalization) and self-report (ED visit) | Hospitalization or ED visit in 12 months | Derivation = 131 (31.9%); Validation = 304 (28.8%) | Derivation = 0.74; Validation = 0.67 | 1. Any of the following conditions: heart disease, diabetes, myocardial infarction, stroke, COPD, cancer 2. 5 or more prescription drugs 3. ED visit or unplanned hospital admission in past 6 months |
| Wu [37] | | Logistic regression Full model approach | 4457 | Leave-one-out cross validation | Medicare beneficiaries aged ≥65 in longitudinal aging study, United States, 2010–2012 | Predictors = 1. survey based (S) 2. claims based (C) 3. survey and claims based (S +C); Outcome = claims data | 1. Any hospital admission within 12 months 2. Preventable hospital admission within 12 months | 1. Any hospital admission = 1046 (21.0%) 2. Preventable hospital admission = 245 (4.5%) | 1. Any hospital admission: • Survey based = 0.67 • Claims based = 0.71 • Combined = 0.72 2. Preventable hospital admission: • Survey based = 0.72 • Claims based = 0.76 • Combined = 0.78 | 1. Frailty status (S)[7] 2. Number of major ADGs[2a] (C, S+C) 3. Number of geriatric risk factors[2c] (C, S+C) |

ADL: activities of daily living, ADG: Aggregated Diagnostic Group, AMTS: Abbreviated Mental Test Score, C: claims based model, CCI: Charlson Comorbidity Index, ED: emergency department, IADL: instrumental activities of daily living, MMSE: Mini-Mental State Examination, NR: not reported, PHN: public health nurse, S: survey assessment based model, C+S: combined survey and claims based model

[1] The Charlson Comorbidity Index incorporates 17 weighted comorbidity conditions. A score of ≥2 is a commonly used cut-point to indicate high comorbidity.

[2] An inclusion criterion for age was not specified. Mean age of the sampled population was 76.1 ± 7.3. (a) Major ADGs refers to 8 major aggregated diagnostic groups assigned by the John Hopkins ACG System, which have very high expected resource use. (b) Hospital dominant conditions were based on diagnoses that are associated with markedly higher probability of future hospitalization. (c) The geriatric risk index was based on the presence of 1 or ≥2 of the 10 geriatric risk factors (i.e. falls, walking difficulty, severe issues with bladder control, absence of fecal control, weight loss, malnutrition, vision impairment, dementia/cognitive impairment, presence of decubitus/pressure ulcers, lack of social support).

[3] Due to multicollinearity between the ADL and IADL limitation variable, two models were developed. In the model with IADL limitation, proxy response was added as predictor. All other variables were identical.

[4] These three variables were separately assessed as number of events in the year before index date and number of events in the year before that.

[5] The screening test was a seven item questionnaire answered by the primary care physician and contained information on limitations in ADLs and IADLs, poor vision, poor hearing, recent unintentional weight loss, use of homecare services, and inadequacy of income.

[6] One of the study participants was aged 62 at inclusion. The geriatric assessment was composed of (a) the Short Form ADL, RAI MDS 2.0 for ADL, (b) the Lawton IADL scale for IADL, (c) the Geriatric Depression Scale for depression, (d) the Mental State Examination for cognition and (e) the mental component score and physical component score of the Short Form-12, a 12-item Health Survey.

[7] Frailty status was categorized as robust, pre-frail and frail, and was based on the five criteria of the Fried frailty phenotype.

**Table 2. Prediction model developed using machine learning techniques.**

| First author + reference | Acronym | Compared algorithms[1] | Population + Setting | Derivation, n | Validation, n | Data used for final model | Outcome | Number of outcome events, n (%) | AUC (95% CI) of best performing algorithm | Features in final model |
|---|---|---|---|---|---|---|---|---|---|---|
| Tarekegn [39] | | SVM ANN RF DT LR GP | Patients in primary care aged ≥65 years, Italy, 2016–2017 | 1) Urgent hospitalization = 1,095,613 2) Preventable hospitalization = 1,095,613 | N/A 10-fold cross-validation procedure | Features and outcome = data from administrative and health databases in the Piedmontese Longitudinal Study | 1) urgent hospitalization 2) preventable hospitalization[2] Horizon = 12 months | Derivation = 1) Urgent hospitalization = 38,918 (3.55%) 2) Preventable hospitalization = 19,072 (1.74%) Validation = N/A | 1) Urgent hospitalization = 0.75 (SVM) 2) Preventable hospitalization = 0.74 (ANN, SVM and LR) | 1) Urgent hospitalization = 34 features[2a] 2) Preventable hospitalization = 33 features[2a] Variable categories: Sociodemographic, medical history, medication, healthcare utilization, functional status |

AUC: area under the curve, CI: confidence interval, DT: decision tree, GP: genetic programming, LR: logistic regression, ML: machine learning, ANN: artificial neural network, RF: random forests; SVM: support vector machine

[1] Algorithms used for feature selection and performance measures, unless stated otherwise.

[2] A definition of preventable hospitalizations was not reported. (a) Ten most important features (equal for urgent and preventable hospitalizations): age, mental disease, poly prescriptions, diseases of the respiratory system, citizenship, non-urgent visit (white code), arthropathy, diseases of the circulatory system, glaucoma. NB These variables were not further specified.

**Table 3.** Variables included in and excluded from the models.

| Category | Variable | Included in final model, N, (%) | Excluded after evaluation, N (%) |
|---|---|---|---|
| **Demographics** | Age | 11 (73%) [28, 31–33, 37–39, 43–46] | 4 (27%) [29, 35, 36, 42] |
| | Sex | 8 (62%) [28, 32, 33, 35, 37, 38, 43, 44] | 5 (38%) [29, 31, 36, 42, 46] |
| | Education | 2 (33%) [33, 45] | 4 (67%) [28, 29, 35, 36] |
| | Race/ethnicity | 2 (40%) [33, 37] | 3 (60%) [28, 29, 31] |
| | Income/SES | 1 (20%) [38] | 4 (80%) [28, 29, 35, 46] |
| | Residential area | 3 (100%) [33, 39, 46] | 0 |
| | Marital status | 1 (33%) [31] | 2 (67%) [36, 45] |
| | Insurance coverage | 2 (50%) [33, 46] | 2 (50%) [29, 31] |
| | Employment | 1 (100%) [35] | 0 |
| **Health status** | Self-rated health | 5 (63%) [28, 29, 35, 40, 42] | 3 (37%) [28, 30, 36] |
| | Mental health | 2 (50%) [34, 41] | 2 (50%) [35, 36] |
| | Physical health | 2 (67%) [34, 41] | 1 (33%) [36] |
| | Use of alcohol or tobacco | 1 (50%) [33] | 1 (50%) [35] |
| **Medical history** | Specific medical diagnoses | 12 (63%) [28–30, 33, 35, 36, 39, 40, 42–45] | 7 (37%) [28, 29, 33, 35, 40, 44, 45] |
| | Multimorbidity | 6 (86%) [31, 32, 36, 37, 44, 46] | 1 (14%) [35] |
| | Sensory impairment | 4 (50%) [32, 33, 37, 38] | 4 (50%) [28, 33, 35, 40] |
| | Cognitive impairment | 5 (83%) [32–34, 37, 40] | 1 (17%) [28] |
| **Health care utilization** | Prior hospitalization | 11 (73%) [28, 29, 31, 35, 36, 38, 40, 42–44, 46] | 4 (27%) [28, 30, 42, 45] |
| | Prior ED visit | 3 (60%) [29, 43, 45] | 2 (20%) [30, 46] |
| | Prior outpatient visits | 2 (40%) [28, 43] | 3 (60%) [28, 30, 42] |
| | Primary care visits | 1 (100%) [31] | 0 |
| | Continuity of care | 0 | 1 (100%) [46] |
| | Receiving homecare | 2 (67%) [38, 45] | 1 (33%) [28] |
| | Previously in LCF | 0 | 3 (100%) [30, 31, 35] |
| | Receiving treatment for specific condition | 1 (50%) [42] | 1 (50%) [31] |
| | Laboratory results | 1 (33%) [35] | 2 (67%) [31, 35] |
| | Barrier to receiving care | 0 | 1 (100%) [29] |
| | Satisfaction with received health care | 0 | 1 (100%) [29] |
| **Medication** | Number of prescription medication | 5 (71%) [36, 38, 39, 44, 46] | 2 (29%) [30, 40] |
| | Use of a specific medication | 2 (67%) [35, 39] | 1 (33%) [35] |
| **Social status** | Caregiver availability | 3 (67%) [28, 41] | 1 (33%) [40] |
| | Lack of social support | 2 (67%) [32, 37] | 1 (33%) [35] |
| | Living arrangement | 1 (14%) [42] | 6 (86%) [28, 29, 33, 35, 36, 40] |
| **Functional status** | ADL | 6 (75%) [29, 33–35, 38, 41] | 2 (25%) [30, 40] |
| | IADL | 3 (50%) [30, 34, 38] | 3 (50%) [29, 30, 35] |
| | Urinary or fecal incontinence | 3 (43%) [32, 33, 37] | 4 (57%) [28, 30, 35, 40] |
| | History of falls | 2 (40%) [32, 37] | 3 (60%) [28, 30, 40] |
| | Mobility | 6 (86%) [30, 35, 39, 40, 42, 46] | 1 (14%) [28] |
| | Malnutrition or weight loss | 3 (100%) [32, 37, 38] | 0 |
| **Other** | Recent stressful event | 0 | 2 (100%) [30, 40] |

*(Continued)*

**Table 3.** (Continued)

| Category | Variable | Included in final model, N, (%) | Excluded after evaluation, N (%) |
|---|---|---|---|
| | Need help to complete survey | 1 (33%) [33] | 2 (67%) [30, 33] |
| | Participation at religious events | 1(100%) [35] | 0 |
| | State of home | 0 | 1 (100%) [42] |

ADL: activities of daily living, ED: emergency department, IADL: instrumental activities of daily living, LCF: long-term care facility, SES: socio-economic status. This table is limited to the information provided in the publications.

frequently included medical diagnoses were cancer (n = 4) [29, 33, 43, 45] and COPD or respiratory problems (n = 4) [33, 39, 44, 45]. Six studies included a multimorbidity measure, either defined as the Charlson Comorbidity Index or a disease count, in the final model [31, 32, 36, 37, 44, 46]. Living arrangement (mostly defined as living alone) was considered for inclusion in seven models [28, 29, 33, 35, 36, 40, 42], and was retained in one model [42]. This model defined living arrangement as living with a spouse.

## Predictive accuracy of the models

Two studies analyzed predictive performance of the same prediction model for two different outcomes [37, 39]. One study did not report its predictive performance [42].

Eighteen studies reported an AUC, ranging from 0.61 to 0.78 after validation. The models published after 2014 tended to perform better; median AUC was 0.72 (range 0.64–0.78) (n = 9), whereas the median AUC from the models in the previous reviews was 0.67 (range 0.61–0.76) (n = 9). Models developed using survey data had median AUC of 0.67 (range 0.61–0.72) (n = 8), the median AUC of models developed with administrative data was 0.73 (range 0.64–0.78) (n = 8). Studies that used both data sources are not included in this count.

The models developed for a specific type of hospitalization (i.e. preventable hospitalization or fall with hospitalization) (n = 3), tended to perform better than the models for all-cause hospitalization (n = 17), with a median AUC of 0.78 (range 0.74–0.78) versus 0.69 (range 0.61–0.76), respectively. The two models that assessed AUCs for both outcomes (i.e. Tarekegn et al. and Wu et al. [37, 39]) were included in calculations of both medians with its corresponding AUC and were thus counted twice.

## Methodological quality

Overall, the methodological quality of included studies was low (Table 4). Risk of bias was either high or unclear in all studies, predominantly due to bias or insufficient reporting in the analysis domain. More specifically, the handling of missing data was not reported or performed inappropriately in ten studies [29, 31, 33, 36, 37, 40–42, 44, 45], eight studies selected predictors based on univariable analyses [30, 31, 33, 35, 40, 42, 43, 45], and five studies solely handled a split-sample procedure for internal validation [28, 30, 31, 33, 45]. Whereas almost all studies (except one [42]) reported model performance in terms of discrimination, only five sufficiently evaluated calibration [28–30, 38, 45]. Four studies only reported results of the Hosmer-Lemeshow test as a single calibration measure [31, 35, 40, 44].

The median EPV was 60 and ranged from 8 [36] to 2003 [45] (n = 15). Two studies reported an EPV <20 [36, 41]. In four studies the EPV could not be computed because data on the number of events or the number of candidate predictors were not reported [30, 33, 34, 42]. The models published after 2014 had a higher EPV (median = 129 (range 27–2003)) than the older models (median = 46 (range 8–64)).

**Table 4. Methodological quality assessment of included prediction models according the recommendations of the PROBAST.**

| First author | Risk of bias | | | | | Applicability | | | Overall | |
|---|---|---|---|---|---|---|---|---|---|---|
| | Participants | Predictors | Outcome | Analysis | EPV[1] | Participants | Predictors | Outcome | ROB | Applicability |
| Boult | - | - | - | + | 48 | - | - | - | + | - |
| Deardorff | - | - | - | ? | 103 | + | - | - | ? | + |
| Freedman | - | - | - | + | NI | - | - | - | + | - |
| Inouye | - | - | - | + | 60 | - | - | - | + | - |
| Kan | + | - | - | ? | 358 | - | - | - | + | - |
| Kim | - | - | - | + | 168 | - | - | - | + | - |
| Kurichi | - | - | - | + | NI | - | - | - | + | - |
| Lin | + | - | - | + | 2003 | + | - | - | + | + |
| Lopez-Aguila | - | - | - | + | 54 | - | - | - | + | - |
| Lyon | - | - | - | + | 44 | - | - | - | + | - |
| Marcusson | - | - | - | + | 87 | - | - | - | + | - |
| Mazzaglia | - | - | - | + | 64 | - | - | - | + | - |
| Mishra | - | - | - | ? | NI | + | - | - | ? | + |
| O'Caoimh | - | - | - | ? | 12 | - | - | - | ? | - |
| Reuben | - | - | - | + | 36 | - | - | - | + | - |
| Roos | - | - | - | + | NI | - | - | - | + | - |
| Shelton | - | - | - | + | 8 | - | - | - | + | - |
| Tarekegn | - | - | -/?[2] | ? | 129 | - | ? | - | ? | ? |
| Wu | - | - | -/+[3] | ? | 27 | - | - | - | ?/+[3] | - |

+: high risk of bias/concern for applicability, -: low risk of bias/concern for applicability,?: unclear risk of bias/concern for applicability. EPV: events per variable, ROB: risk of bias, NI: no information (i.e. either number of events or number of candidate predictors was not reported)

[1] For studies where multiple outcomes were assessed, only the lowest number of events per variable per study is reported.

[2] For the outcome preventable hospitalization, no definition was reported, ROB was therefore evaluated as unclear. For the outcome acute hospital admission, ROB in this domain was low.

[3] ROB was low for the outcome any inpatient hospital admission. ROB was high for the outcome preventable hospital admissions, since predictors were included in the outcome definition. Overall ROB was therefore unclear and high, respectively.

Concern for applicability was high in three studies, because the study population or study outcome did not fully match the review question: one study only included older adults with a sensory impairment [29], one study excluded older adults with a hospital admission <6 months prior to the index date [45], and one study evaluated preventable hospital admissions as only outcome [46].

## Discussion

This systematic review identified 19 prediction models to predict unplanned hospital admissions in community-dwelling older adults. With our search strategy we built on a review by Wallace et al. on the same topic, however focusing the study population to adults aged 65 years and over. In total we identified 19 prediction models, of which the current review added 10 new prediction models that were not included in the previous reviews. The new models had higher predictive accuracy than the older models. This might be explained by the fact that new models had larger samples of the development cohort and also higher EPVs than the older models. Both are recommended by the TRIPOD guidelines, published in 2015 [16], to improve predictive accuracy and methodological quality. Moreover, the new models used administrative or clinical record data more often for the development of their model. Consistent with

Wallace et al., we found that models developed using administrative or clinical record data had higher predictive accuracy than those developed using self-report data. Of the 10 new prediction models, eight used administrative data for development of their model.

To potentially improve predictive accuracy, Wallace et al. suggested to consider nonmedical factors (e.g. social support and functional status) [13]. Despite this recommendation, these variables were rarely evaluated for inclusion in the latest studies. We found that predictors most frequently included in the final models were medical diagnoses (specifically heart disease), prior hospitalizations, age, and sex, which is in line with Wallace's findings. These risk factors seem to have more impact in the prediction of unplanned admissions than nonmedical factors, considering the relatively high beta-coefficients of these variables in most models (data not shown). Also, chronic diseases and health care use variables are probably more readily available in large routine care data, whereas nonmedical factors are rarely assessed in a systematic way.

Overall, reporting of methodology and findings was often inappropriate or lacked relevant information, risk of bias was therefore either unclear or high in all models. Moreover, despite the publication of the TRIPOD guidelines in 2015, only one [29] out of seven studies published after 2015 reported their study according to the TRIPOD checklist. The majority of studies showed high risk of bias in the analysis domain. Mainly because of univariable analyses as selection method or inappropriate handling of missing data.

## Strengths and limitations

The aging population across the globe and increasing interest in personalized medicine makes this review topical. We added a substantial number of prediction models to the previous systematic reviews on this topic. Furthermore, we conducted a thorough search strategy using a validated search filter and assessed data using tools specifically designed for systematic reviews of prognostic studies. However, there are some limitations. First, care must be taken with directly comparing the prediction models because of heterogeneity in study characteristics (e.g. study populations, and selection of candidate predictors) and study outcomes. Since models perform differently in other populations, comparison of predictive performance can only be performed when these models are validated in the same sample. Further, by limiting our inclusion criteria to participants aged 65 and over, we excluded potential prediction models developed in participants with younger age. For example, the DIVERT scale, a tool to predict emergency department visits, was developed in home care clients aged ≥50 years. Even though reported AUCs are a little over 0.6 after geographical validation, targeted application of the risk score has shown its clinical added value for cardiorespiratory management and reduction of hospitalizations in home care recipients [47]. Last, while in principle CHARMS and PRO-BAST are relevant for prediction model studies using ML, they predominantly focus on regression-based modelling and some unique aspects of ML methods are not captured [48]. This complicated the critical appraisal of the ML study and therefore risk of bias was unclear. Necessity for guidelines for reporting and critical appraisal of prediction model studies using ML has been addressed and PROBAST-ML (as well as TRIPOD-ML) has been announced [48]. Until then, it is recommended to use TRIPOD, CHARMS and PROBAST as benchmark for the development of prediction model studies rather than none [49].

## Implications for future research

Our findings provide a proper basis of prediction models on hospitalizations in older people. Knowing that prediction models often perform worse in new populations, external validation studies are needed to assess generalizability across different countries and healthcare systems.

Moreover, models that underperform in external samples should not be discarded and studies should assess the possibility of updating existing models by recalibrating, adjusting weights or considering additional predictors [50, 51]. This way, data of the original development model is not wasted. However, updating of a prediction model is only recommended provided that the initial model was appropriately developed and demonstrated promising accuracy [51]. Most prediction models in this review are poorly reported and all are at either high or unclear risk of bias, which makes updating of the existing models more complicated and we therefore cannot recommend one specific model.

Moreover, while recalibration and adjusting weights only affect a model's calibration, adding (previously missed) important predictors should be considered to improve a model's discrimination [51]. As mentioned above, nonmedical factors remain under researched in the prediction of hospital admissions in older adults. Taking into account the influence of nonmedical factors on unscheduled secondary care use [52, 53], these variables may contribute to a better discriminative ability of the model.

Last, for both development studies and validation studies we advise to fully report all modelling steps and analysis in sufficient detail according to the TRIPOD guidelines [16]. The TRIPOD guidelines have been developed to improve the reporting of studies developing, validating, and updating prognostic models and to maximize transparency and reproducibility. More specifically, for example, predictive performance should not only be evaluated in terms of discrimination, but also in terms of calibration. Regarding calibration, it is recommended to include a calibration plot or table in addition to the p-value of the Hosmer-Lemeshow test. Furthermore, variables or participants with missing data should not simply be omitted, multiple imputation is recommended as the preferred method for handling of missing data to decrease bias [26].

## Implications for future practice

Our study found that the models to predict preventable hospitalizations tended to have better predictive ability than models for all-cause hospitalizations. Preventable admissions reflect admissions for conditions that could have been managed with timely and effective treatment by outpatient primary care (e.g. pneumonia, congestive heart failure, and COPD, often also referred to as ambulatory care sensitive conditions (ACSCs)) [54]. Interventions targeted at older adults with ACSCs provide a window of opportunity for prevention of admissions. Possibly even more so if targeted at persons with additional important risk factors (e.g. recent hospitalization, polypharmacy and/or multimorbidity). In consequence, reduction of the incidence of preventable admissions could substantially lower healthcare costs, and improve health outcomes and older adult's quality of life [11].

There is however limited evidence for effective preventive interventions to reduce preventable admissions in general [54]. High continuity of care with a general practitioner is associated with lower rates of hospital admissions [55]. Furthermore, several targeted interventions have shown to be effective in patients with specific diseases, such as self-management in patients with COPD and heart failure, and telemedicine in patients with heart failure [11]. Focusing on these targeted interventions may have a beneficial impact on the reduction of hospital admissions in community-dwelling older adults [54].

## Conclusion

The prediction models developed to predict preventable hospitalizations tended to perform better than models predicting all-cause hospitalizations. Focusing on enhancing primary care

management of conditions related to these preventable admissions may have a beneficial effect on health care quality.

To improve predictive accuracy of prediction models the use of administrative data sources is recommended as well as incorporation of important variables, i.e. age, prior hospitalization and multimorbidity. The impact of nonmedical factors remains unresearched. Moreover, future researchers are recommended to follow the TRIPOD guidelines for prediction model studies, as methodological quality of reporting and analyses of the included studies was low.

## Supporting information

**S1 Checklist. PRISMA checklist.**
(PDF)

**S1 File. Full search strategies.**
(PDF)

**S2 File. CHARMS and PROBAST forms.**
(PDF)

## Acknowledgments

The authors thank Stichting Preventie, Vroegdiagnostiek en eHealth for their support.

## Author Contributions

**Conceptualization:** Jet H. Klunder, Karlijn J. Joling, Otto R. Maarsingh, Hein P. J. van Hout.

**Data curation:** Jet H. Klunder, Sofie L. Panneman.

**Formal analysis:** Jet H. Klunder, Sofie L. Panneman.

**Investigation:** Jet H. Klunder, Sofie L. Panneman.

**Methodology:** Jet H. Klunder, Emma Wallace, Ralph de Vries, Karlijn J. Joling, Otto R. Maarsingh, Hein P. J. van Hout.

**Supervision:** Karlijn J. Joling, Otto R. Maarsingh, Hein P. J. van Hout.

**Writing – original draft:** Jet H. Klunder, Sofie L. Panneman.

**Writing – review & editing:** Jet H. Klunder, Emma Wallace, Ralph de Vries, Karlijn J. Joling, Otto R. Maarsingh, Hein P. J. van Hout.

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
