## [Decision Letter · Decision Letter 0]

2 Mar 2022

PONE-D-21-38410Risk prediction models for the prediction of unplanned hospital admissions or emergency department visits in community-dwelling older adults: a systematic review.PLOS ONE

Dear Dr. Klunder,

Thank you for submitting your manuscript to PLOS ONE. After careful consideration, we feel that it has merit but does not fully meet PLOS ONE’s publication criteria as it currently stands. Therefore, we invite you to submit a revised version of the manuscript that addresses the points raised during the review process.

ACADEMIC EDITOR:

We now have 3 reviews available of your manuscript. You will see that the reviewers were quite positive about this research but did raise a number of suggestions to improve the clarity of the manuscript - some related to the structuring of the manuscript and other focused on providing more detail on certain decisions. As well, you will also see that the reviewers raised some concerns about the discussion of machine learning techniques vs regression and are generally looking for a more nuanced take on this issue.

We look forward to receiving your revised manuscript.

Kind regards,

Andrea Gruneir

Academic Editor

PLOS ONE

Journal Requirements:

Reviewers' comments:

Reviewer's Responses to Questions

**Comments to the Author**

1. Is the manuscript technically sound, and do the data support the conclusions?

Reviewer #1: Yes

Reviewer #2: Yes

Reviewer #3: Partly

2. Has the statistical analysis been performed appropriately and rigorously? 

Reviewer #1: N/A

Reviewer #2: N/A

Reviewer #3: N/A

3. Have the authors made all data underlying the findings in their manuscript fully available?

Reviewer #1: Yes

Reviewer #2: Yes

Reviewer #3: Yes

4. Is the manuscript presented in an intelligible fashion and written in standard English?

Reviewer #1: Yes

Reviewer #2: Yes

Reviewer #3: No

5. Review Comments to the Author

Reviewer #1: An interesting systematic review on risk prediction models for hospitalizations among community-dwelling older adults. The review looks to be well-conducted and the conclusions soundly based in the results. I appreciate the focus on the methodological quality of the studies in addition to their performance. I have a few suggestions that I believe well aid in the clarity of their manuscript

Abstract

1. Line 25: In the Abstract the inclusion criteria is that models had to be intended to be used in a “primary care setting” whereas in the Methods section it is “general practice or community care”. This may be a jurisdictional difference, but I would not typically consider community care to be part of primary care so do not see these statements as equivalent. Can the authors align the definitions?

2. Line 27: Do the authors mean “quality” assessment?

3. Line 31: I believe that “sex or gender” would be more accurate according to the data in Table 2

Introduction

4. Lines 60-63. I might be misunderstanding this sentence, but it sounds like the authors are saying that regression models at more at risk of overfitting than machine learning models, which is not typically true. Machine learning models are much more complex than regression models. This complexity leads to both a theoretical benefit for predictive performance as well as an increased risk of overfitting. Some of the papers that the authors have cited [13 – for example] discuss this.

Methods

5. Line 108. I’m confused about the scope and justification of this exclusion. I can understand wanting to excluding studies done in disease-specific populations. But I am not grasping why studies among patients with cognitive impairments were specifically included when studies done in patients with heart failure, for example, presumably were excluded. Could the authors provide more detail and justification?

6. Line 132: Could the authors briefly list the domains of high heterogeneity in this sentence?

Results

7. Line 203: “sex or gender” is likely more accurate

Table 3

8. Presentation of the data in an N(%) format would be more informative than listing the reference numbers

Discussion

9. Feature/variable selection is a controversial and complex topic and I think the authors could benefit from more nuance in their discussion. For example, backwards selection is clearly superior to univariable screening, but it still comes with its own challenges. The use of any automated selection model (as noted in [14 and 15]) bears risks and p-value based approaches in particular lack justification. Could the authors expand this section, comment on other available methods ,i.e. LASSO, other methods as detailed in https://doi.org/10.1016/j.jclinepi.2015.10.002 and https://doi.org/10.1016/j.ijmedinf.2018.05.006 and comment that some ML methods have feature/variable selection incorporated into their algorithms.

Reviewer #2: The authors have presented an updated systematic review in this paper. The study is interesting and the approach is adequately robust. My specific comments are given below.

1. Is there a specific rationale for the chosen time frame for searching literature (2013-2021)?

2. Also, is there a specific reason for restricting the participants to adults > 65 years of age?

3. Time frame ranges from 7 days to 4 years. Was there any discernible temporal decay in the predictive performance across the models over this relatively longer time window? In other words, did those models predicting a shorter time span perform better than those predicting longer time spans?

4. The study has found that models developed to predict preventable hospitalizations had better predictive performance than models predicting hospitalizations in general. I think the authors should elaborate further on the clinical implications of this important finding.

5. Machine learning/deep learning-based models are quite different from traditional statistical models in a number of ways. The inclusion of a large number of variables in ML/DL models is, in fact, not an issue and modelling in a high-dimensional space is permissible with ML/DL. Therefore, using the guidelines (TRIPOD/CHARMS/PROBAST) geared to assessing classic predictive models for ML/DL models may not be ideal. Authors should discuss the implications of this and likely limitations.

6. Authors have presented the different variables included in each model. What predictors were actually found to be important in these predictive models? What predictors were statistically significant in classic predictive models and what variables emerged as important in ML models? For instance, ML uses techniques such as variable importance metrics and Shapley additives to gauge predictor importance.

7. Suggested to include eligibility criteria in a standard PICOTS table.

8. It would be important to describe in detail what additional and novel findings emerged from this SR, compared to the two previous SR on the same domain.

9. Both split-sample validation and cross-validation have their limitations. External validation is a great way to assess the robustness and generalizability of predictive models. How many of these models were externally validated?

Reviewer #3: First of all I would like to thank and congratulate the authors on their work. The topic of this systematic review is very interesting and important. The manuscript lacks however, structure and does not always read well. See my suggestions and questions below to improve this work.

BACKGROUND

Use of “risk” prediction modeling is a confusing and uncommon terminology. Recommend to use solely prediction model. This will probably improve the readability of the manuscript.

A very large proportion of the introduction is being used to describe prediction models, big data and machine learning in general. This does not read well, and does not add very much value to the topic of this systematic review. I would recommend to explain more about the “burden” of older patients at the emergency department. For example, how many times are patients admitted to and ED?; What are the reasons that they visit the ED? Are these reasons preventable? This will highlight the importance of this research. Additionally, I would also recommend to focus on the effects of ED admission and hospitalization on the elderly, such as the loss of functionality, risk of delirium during admission, psychological effects etc. Additionally, the authors described that with an effective primary care intervention, healthcare costs will decrease. I suggest to add details about how identification of these elderly can improve the work of physicians on how they can deliver more qualitative and effective healthcare.

METHODS

My major concern is the following: in the introduction and methods it is explained that previous reviews included studies focusing on ED admission and case-finding instruments. However, inclusion for this study was limited to studies from 2013 onwards, despite focusing on ED admission and unplanned hospitalization. The reason why the authors chose this specific inclusion year is confusing, as the previous performed reviews do not fully cover the research question of this systematic review. Could the authors explain, how and why this decision was made.

Inclusion criteria one, two and four seem obvious. However I think inclusion criteria three and five need more explanation. In general a question to authors: why were only validated prediction models included in this study? With the PROBAST tool, the models are also scored on “validation”. I do not see why development studies are not included.

In regards to inclusion criteria five: how do prediction models being used at the ED differ from the ones being used at a primary care facility?

Textual comments:

Methods section reads cloudy and could be more straightforward. For example: [1] Since these systematic reviews identified the same risk prediction models, we decided to limit publication dates from August 2013 through January 2021, which has some overlap with the searches of these reviews. To give a complete overview, we will also include the studies found in the previous reviews. The references of the identified articles were searched for relevant publications. I would suggest to rephrase to: [1] To provide a complete overview of available prediction models our search was restricted to August 2013 through January 2021. The models described in the previous reviews were also included in this systematic review.

Textual comments:

[2] After extraction of data, the Prediction model Risk of Bias Assessment Tool (PROBAST; see Appendix B) was used to assess risk of bias and applicability of the predictive models. Concern for applicability addresses whether the primary study matches the review question. I would suggest to rephrase to: [2] The Prediction model Risk of Bias Assessment Tool was used to assess risk of bias and applicability, of which the latter addresses whether the primary study matches the review question.

I would suggest to change the structure of the methods section and shorten in. Combine sections search strategy, study selection and data extraction. Make a new subheading with model performance including de explanation about AUC and EPV and lastly discuss the PROBAST tool. The PROBAST tool is explained very extensively. I would recommend to remove details to supplements or just refer to original PROBAST article. Also describe that regression models and machine learning models will be described separately.

RESULTS

A question for authors: Did all of the included study focus on developing only 1 prediction model? Because in these kind of studies sometimes multiple models are developed/ validated and compared to each other. Is 22 studies equivalent to 22 unique prediction models?

Textual comments

Same as the methods. Results can be pointed out more straightforward. See examples below.

Line 162/168: A flow diagram of the search strategy and selection process is presented in Figure 1, can be removed. Data extracted from the studies can be found in Table 1 and Table 2. Suggest to change it to: The literature searches yielded a total of 16,098 citations (Figure 1.). Tables and figures do not a notification, referring is the standard.

Line 164. Additionally, twenty-three articles were identified through other sources. What are these other sources? This was not mentioned in the methods.

Line 165: In addition to 10 studies included in the previously published systematic reviews, 12 new studies met all inclusion criteria, which makes a total of 22 unique risk prediction models. Rephrase: Full texts were retrieved for 170 studies of which 12 met all inclusion criteria. Additionally, a total of 10 studies were included from the previously published systematic reviews.

I would suggest to refer to “prediction model” instead of “study” in the results sections. For example line 171: Thirteen studies included participants aged ≥65 172 years[29, 33, 34, 36-40, 44, 46-49], the remaining studies used a higher age as inclusion criterion with. Rephrase to: Thirteen prediction models included……..

When describing results, try to hold on to the structure in the methods. The EPV can be described in the “predictive accuracy” section.

Avoid using question marks in tables and figures. I would suggest to use NA or a color scheme, for example, high risk= red, low risk= green, unclear= purple.

DISCUSSION

I do not understand why the difference between machine learning and logistic regression models is discussed prominently in this article. In order to say whether one technique is superior to the other, you should validate both models in the identical population. In line 317 the authors state the machine learning techniques are not superior and in in line 328 the authors state that a fair comparison is not possible. Please be consequent in conclusions.

The discussion includes a lot of repetition of the results. Conclusions is solely based on the development of more prediction models. In the introduction the authors describe that the eventual goal is to develop a care management program to avoid these admission. The authors should highlight, how they could use these models to develop such a program. For example; are the variables in these model, standard measurements in a primary care facility?

If the conclusion is only developing more models, the authors should describe how to accomplish this. Which data source to use, which variables should be included, which modeling technique etc.

Textual comments:

Line 293: Twelve risk models were added to the existing evidence. This does not read well. Suggest to remove this sentence.

Line 293-295: The recommendation of using nonmedical factors is never mentioned in the manuscript. This conclusion comes a bit out of the blue. I would recommend to make a more general conclusion, on the results that the authors did find (e.g. quality of models, performance of models etc.).

6. PLOS authors have the option to publish the peer review history of their article (what does this mean?). If published, this will include your full peer review and any attached files.

Reviewer #1: No

Reviewer #2: No

Reviewer #3: No

---

## [Author Response · Author response to Decision Letter 0]

29 May 2022

Amsterdam, May 1st 2022

Dear editor,

We would like to thank you and the reviewers for thoroughly reviewing our manuscript. We have rigorously revised our manuscript and believe its quality has improved. We have addressed each comment point by point below. We have marked important changes in the revised manuscript with track changes and have referred to the page and line numbers in the commentary below.  

Reviewer #1: An interesting systematic review on risk prediction models for hospitalizations among community-dwelling older adults. The review looks to be well-conducted and the conclusions soundly based in the results. I appreciate the focus on the methodological quality of the studies in addition to their performance. I have a few suggestions that I believe well aid in the clarity of their manuscript

Abstract

1. Line 25: In the Abstract the inclusion criteria is that models had to be intended to be used in a “primary care setting” whereas in the Methods section it is “general practice or community care”. This may be a jurisdictional difference, but I would not typically consider community care to be part of primary care so do not see these statements as equivalent. Can the authors align the definitions?

We agree these terms may be confusing and we thank the reviewer for highlighting this contradiction. Primary care focuses on physical, mental and social health issues, it comprises care mainly performed in general practice and home care. Community care addresses a wider aspect of wellbeing and also focuses on social problems people may experience, such as housing problems. The aim of this review is to find risk prediction models that can be used by either general practitioners or home care professionals, therefore primary care is the most suitable definition. We have changed the definition in both abstract and methods accordingly. (page 2, line 24; page 4, line 104)

2. Line 27: Do the authors mean “quality” assessment?

We do indeed. We’ve changed this. (page 2, line 26)

3. Line 31: I believe that “sex or gender” would be more accurate according to the data in Table 2

We assume all studies refer to the biological characteristics of males and females, which means sex would be the appropriate term. However some included studies use the term gender and do not describe their definition, therefore we changed ‘gender’ into ‘sex or gender’ according to the reviewer’s suggestion. (p2, line 29)

Introduction

4. Lines 60-63. I might be misunderstanding this sentence, but it sounds like the authors are saying that regression models at more at risk of overfitting than machine learning models, which is not typically true. Machine learning models are much more complex than regression models. This complexity leads to both a theoretical benefit for predictive performance as well as an increased risk of overfitting. Some of the papers that the authors have cited [13 – for example] discuss this.

We thank the reviewer for their comment. We could indeed write a whole article on the benefits and disadvantages regression and ML models. In light of the reviewer’s comment and the exclusion of some studies after this peer review, we have decided put less emphasis on the differences between regression and ML in our review, and we have therefore removed this paragraph from our introduction. 

Methods

5. Line 108. I’m confused about the scope and justification of this exclusion. I can understand wanting to excluding studies done in disease-specific populations. But I am not grasping why studies among patients with cognitive impairments were specifically included when studies done in patients with heart failure, for example, presumably were excluded. Could the authors provide more detail and justification?

We agree with the reviewer that the argument to exclude prediction models developed in specific populations except for community-dwelling older adults with cognitive impairments is disputable. We have therefore adjusted this criterion in our methods (page 5, lines 111-113) and excluded the two corresponding studies (i.e. Tsang et al.[1] and Maust et al.[2]) from this review. 

6. Line 132: Could the authors briefly list the domains of high heterogeneity in this sentence?

This sentence refers to the impossibility to perform a meta-analysis due to high heterogeneity among the included studies. Our study describes prediction model development studies, these are developed using different statistical methodologies (i.e. different regression analysis methods and ML). A meta-analysis summarizes the estimates of model discrimination and calibration, however due to the wide variation in, among others, outcomes (i.e. (preventable) hospital admissions or both) and prediction horizons (days to years) a quantified average performance in terms of discrimination or calibration would be uninterpretable. 

Results

7. Line 203: “sex or gender” is likely more accurate

We have changed this according to their suggestion. (page 8, line 178)

Table 3

8. Presentation of the data in an N(%) format would be more informative than listing the reference numbers

We thank the reviewer for their suggestion. We agree with the reviewer that presentation of percentages in this table is of added value, and we have therefore included these to Table 3 (page 17). 

Discussion

9. Feature/variable selection is a controversial and complex topic and I think the authors could benefit from more nuance in their discussion. For example, backwards selection is clearly superior to univariable screening, but it still comes with its own challenges. The use of any automated selection model (as noted in [14 and 15]) bears risks and p-value based approaches in particular lack justification. Could the authors expand this section, comment on other available methods ,i.e. LASSO, other methods as detailed in https://doi.org/10.1016/j.jclinepi.2015.10.002 and https://doi.org/10.1016/j.ijmedinf.2018.05.006 and comment that some ML methods have feature/variable selection incorporated into their algorithms.

We agree with the reviewer that this section is somewhat short-sighted and we have nuanced this section (page 21, lines 337-341). As stated by the reviewer, variable selection is complex and controversial and many methods for variable selection exist. However, for this review we strictly followed the recommendations of the TRIPOD.[3] The TRIPOD is currently considered as the state-of-the-art reporting guideline for prediction model studies and systematic reviews on this topic.[4, 5] According to the TRIPOD, backward elimination is generally preferred if automated predictor selection procedures are used. While some of TRIPOD’s recommendations may offer room for in-depth discussion about the different predictor selection methods, we believe that this is beyond the scope of this paper, especially considering the exclusion of 2/3 ML models after revision. If the editor would like us to elaborate on this issue further, we would like to know and we will gladly accommodate their request. 

 

Reviewer #2: The authors have presented an updated systematic review in this paper. The study is interesting and the approach is adequately robust. My specific comments are given below.

1. Is there a specific rationale for the chosen time frame for searching literature (2013-2021)?

The aim of this review was to update the systematic review of Wallace et al. and in addition, focus on prediction models in adults aged 65 and over. Wallace et al. have performed a thorough literature search with identical inclusion criteria as our study (except for the age criterion). Since Wallace’s last updated search was in February 2014 we decided to overlap our search with 6 months, in case any records were added retrospectively (e.g. to correct an error in indexing) and were not detected by Wallace’s search. 

2. Also, is there a specific reason for restricting the participants to adults > 65 years of age?

Older adults have greater vulnerability to acute stress than younger individuals due to age-related diminution of physiologic reserves. Moreover, older adults tend to have more comorbid illnesses and disability. Older adults are more at risk for hospitalization and require more support after discharge than adults in middle age (45 to 64 years).[6] Prevention of hospital admissions in the older age group seems therefore more important on a community-level. There is however no clear age that defines an older adult. We have chosen the most conventional definition of older adults: people aged 65 and over.[7] 

We have revised our introduction, putting more focus on the importance of identifying older adults at risk for hospitalizations. 

3. Time frame ranges from 7 days to 4 years. Was there any discernible temporal decay in the predictive performance across the models over this relatively longer time window? In other words, did those models predicting a shorter time span perform better than those predicting longer time spans?

Most studies handled a prediction horizon of 12 months (n=12, 63%). The predictive performance of these models, excluding the ones that predicted preventable hospitalizations or fall with hospitalization, ranged between AUC 0.61 – 0.76 (median AUC = 0.70). Two models were developed to predict a shorter time span, i.e. 6 months (AUC 0.72), and 4.5 months (AUC 0.63). So there was no remarkable superiority of the prediction models with shorter prediction horizon in terms of predictive performance. On the other hand, four studies handled a prediction horizon of more than 12 months (i.e. 15 months to 4 years). These models showed worse predictive ability (median AUC = 0.67, range 0.61 – 0.69) than the models predicting a time span of 12 months. This suggests predictions over a time span of more than 12 months become less accurate. In addition, it is questionable whether predictions of hospital admissions within an interval of multiple years are of clinical added value to clinicians. However, their low accuracy could also be due to the fact that these studies used survey data, whereas models using administrative data tended to perform better. We did therefore not incorporate this finding in the description of our results. 

4. The study has found that models developed to predict preventable hospitalizations had better predictive performance than models predicting hospitalizations in general. I think the authors should elaborate further on the clinical implications of this important finding.

We thank the reviewer for their suggestion and have elaborated on this finding and its clinical implications (page 21-22, lines 353-370)

5. Machine learning/deep learning-based models are quite different from traditional statistical models in a number of ways. The inclusion of a large number of variables in ML/DL models is, in fact, not an issue and modelling in a high-dimensional space is permissible with ML/DL. Therefore, using the guidelines (TRIPOD/CHARMS/PROBAST) geared to assessing classic predictive models for ML/DL models may not be ideal. Authors should discuss the implications of this and likely limitations.

On page 20, in line 321 and further we discuss the fact that TRIPOD, CHARMS and PROBAST are originally designed for all types of prediction modelling studies, however their focus is indeed on regression-based prediction models. Nonetheless, all TRIPOD items are applicable for ML models. For PROBAST however, two signaling questions might be less relevant (i.e. selection of predictors based on univariable analysis and reporting of weighted estimates in the final model) and more signaling questions, e.g. related to data generation and feature selection, might be necessary. Hence, risk of bias for the ML model in our review was signed as unclear. ML versions of these checklists are under development, until then it is recommended to use TRIPOD, CHARMS and PROBAST as benchmark rather than none. We have added the latter in the manuscript (see underlined sentence below):

Necessity for guidelines for reporting and critical appraisal of prediction model studies using ML has been addressed and PROBAST-ML (as well as TRIPOD-ML) has been announced.[8] Until then, it is recommended to use TRIPOD, CHARMS and PROBAST as benchmark for the development of prediction model studies rather than none.[9] 

6. Authors have presented the different variables included in each model. What predictors were actually found to be important in these predictive models? What predictors were statistically significant in classic predictive models and what variables emerged as important in ML models? For instance, ML uses techniques such as variable importance metrics and Shapley additives to gauge predictor importance.

Thanks to the reviewer for their suggestion. We have presented all variables that were statistically significant and thus included in the final models in Table 3. Because of other comments on the difference between classic regression models and ML models, we decided to put less emphasis on the differences between these models. To answer the first question; because every study handled predictors differently (e.g. categorical or continuous analysis of the variable age) and used different numbers of predictors and different predictor selection methods, no quantified conclusions can be drawn on which predictors were most important. In general, previous admissions, high age, multimorbidity, polypharmacy and heart disease were most frequently included in the final models with in most cases high beta-coefficients. We added this to our conclusions. (page 19, line 278-282; page 21, line 359-361)

7. Suggested to include eligibility criteria in a standard PICOTS table.

We have adapted the inclusion criteria in the manuscript according to the PICOTS format as suggested by Debray et al.[4] (page 4, line 95-104)

8. It would be important to describe in detail what additional and novel findings emerged from this SR, compared to the two previous SR on the same domain.

We thank the reviewer for their suggestion. We have now described our results in the discussion in light of the findings by the previous reviews. (page 19, lines 276-297)

9. Both split-sample validation and cross-validation have their limitations. External validation is a great way to assess the robustness and generalizability of predictive models. How many of these models were externally validated?

The models that were externally validated within the same study are reported in Table 1 (n=3, i.e. Marcusson, Mazzaglia, and Shelton). Unfortunately, very few models were externally validated by other researchers. We have not incorporated these in our studies since our primary aim was to summarize predictive models that have been developed to predict unplanned hospital admissions to this date. 

Reviewer #3: First of all I would like to thank and congratulate the authors on their work. The topic of this systematic review is very interesting and important. The manuscript lacks however, structure and does not always read well. See my suggestions and questions below to improve this work.

BACKGROUND

Use of “risk” prediction modeling is a confusing and uncommon terminology. Recommend to use solely prediction model. This will probably improve the readability of the manuscript.

We thank the reviewer for their suggestion and we have changed this accordingly in the manuscript.

A very large proportion of the introduction is being used to describe prediction models, big data and machine learning in general. This does not read well, and does not add very much value to the topic of this systematic review. I would recommend to explain more about the “burden” of older patients at the emergency department. For example, how many times are patients admitted to and ED?; What are the reasons that they visit the ED? Are these reasons preventable? This will highlight the importance of this research. Additionally, I would also recommend to focus on the effects of ED admission and hospitalization on the elderly, such as the loss of functionality, risk of delirium during admission, psychological effects etc. Additionally, the authors described that with an effective primary care intervention, healthcare costs will decrease. I suggest to add details about how identification of these elderly can improve the work of physicians on how they can deliver more qualitative and effective healthcare.

We thank the reviewer for their suggestion. We have revised the introduction according to their suggestion by omitting a large part of the paragraph on machine learning and describing in more detail about the relevance of the topic of our review. (page 3, lines 40-55)

METHODS

My major concern is the following: in the introduction and methods it is explained that previous reviews included studies focusing on ED admission and case-finding instruments. However, inclusion for this study was limited to studies from 2013 onwards, despite focusing on ED admission and unplanned hospitalization. The reason why the authors chose this specific inclusion year is confusing, as the previous performed reviews do not fully cover the research question of this systematic review. Could the authors explain, how and why this decision was made.

We understand the reviewer’s concern. With this review we build on the review of Wallace et al. who performed a thorough review in 2014 (20,666 records), for this review our focus was on older adults because of reasons (mentioned in the introduction). Wallace et al. included risk prediction models for hospital admissions or combined endpoints such as hospital admission or ED visits, but did not include prediction models predicting ED visits as single outcome. We agree that this is a gap in our search strategy. We therefore amended our inclusion criterion to hospital admissions only or a combined endpoint of hospital admissions and ED visits. (page 4, line 100-103)

This way our inclusion criteria are identical to Wallace et al., except for the restriction to the older population. 

In consequence, the modification of this inclusion criterion led to the exclusion of one study (i.e. Veyron et al.[10]) in the original version of this review.

Inclusion criteria one, two and four seem obvious. However I think inclusion criteria three and five need more explanation. In general a question to authors: why were only validated prediction models included in this study? With the PROBAST tool, the models are also scored on “validation”. I do not see why development studies are not included.

We thank the reviewer for their comment. Internal validation is considered as a basic procedure in prediction model development studies (in general, but especially for studies with small sample size and/or low EPV). According to the TRIPOD, development studies are defined as development of a prediction model without validation in other participant data, but with inclusion of some form of resampling technique (in other words; including internal validation). Development studies without any form of validation are at high risk of overfitting and thus, according to PROBAST, in principle at high risk of bias. To describe studies that might be useful in daily practice, we only included studies in which overfitting was already accounted for through the execution of any internal validation procedure.

In regards to inclusion criteria five: how do prediction models being used at the ED differ from the ones being used at a primary care facility?

Risk assessment does differ between these settings. Patients already admitted to the ED have higher a priori probability to be admitted to hospital than patients being at home, when risk assessment is performed. Therefore, predictive performance of prediction models in these two settings cannot be compared. Consequently, other variables are selected for inclusion in the models. For instance, the APOP screener [11] includes a variable whether the patient arrived by ambulance. This question can obviously not be answered when this model is used in a primary care setting. 

Textual comments:

Methods section reads cloudy and could be more straightforward. For example: [1] Since these systematic reviews identified the same risk prediction models, we decided to limit publication dates from August 2013 through January 2021, which has some overlap with the searches of these reviews. To give a complete overview, we will also include the studies found in the previous reviews. The references of the identified articles were searched for relevant publications. I would suggest to rephrase to: [1] To provide a complete overview of available prediction models our search was restricted to August 2013 through January 2021. The models described in the previous reviews were also included in this systematic review.

We thank the reviewer for their suggestion, we have changed this section accordingly. (page 4, lines 88-91)

Textual comments:

[2] After extraction of data, the Prediction model Risk of Bias Assessment Tool (PROBAST; see Appendix B) was used to assess risk of bias and applicability of the predictive models. Concern for applicability addresses whether the primary study matches the review question. I would suggest to rephrase to: [2] The Prediction model Risk of Bias Assessment Tool was used to assess risk of bias and applicability, of which the latter addresses whether the primary study matches the review question.

We have changed the sentence accordingly to the reviewer’s suggestion. (page 5, lines 135-137)

I would suggest to change the structure of the methods section and shorten in. Combine sections search strategy, study selection and data extraction. Make a new subheading with model performance including de explanation about AUC and EPV and lastly discuss the PROBAST tool. The PROBAST tool is explained very extensively. I would recommend to remove details to supplements or just refer to original PROBAST article. Also describe that regression models and machine learning models will be described separately.

We thank the reviewer for their suggestion. We have shortened the methods section and removed the elaboration on the PROBAST and referred to the original PROBAST article (page 5, lines 135-139). We have also added a sentence about the separate description of regression models and ML models (page 5, lines 128-129).

RESULTS

A question for authors: Did all of the included study focus on developing only 1 prediction model? Because in these kind of studies sometimes multiple models are developed/ validated and compared to each other. Is 22 studies equivalent to 22 unique prediction models?

The majority of studies developed one prediction model, but some indeed presented more models. We presented data of all reported models in Tables 1 and 2. In general we counted one model per study, with the only exception for the comparison of predictive performance between all-cause hospitalizations and preventable hospitalizations. We have added a sentence in the results section to clarify this. (page 8, lines 191-195) We have amended the use of the words study and model, when necessary.

The way we handled the different presentation of models is described in detail below:

• In case multiple validation procedures were performed (i.e. Lyon and Marcusson) we only reported the highest AUC in the description of our results. In Table 1 all AUCs are reported.

• Three studies (i.e. Kan, Reuben, and Wu) used (and combined) multiple data sources to develop multiple models (e.g. survey data and electronic record data). For clarity reasons, we evaluated and discussed the studies as one model, because study characteristics and variables in the final model were identical. For the description of predictive accuracies we only counted the model with the highest AUC. All AUCs are reported in Table 1.

• Three studies (i.e. Mishra, Tarekegn, and Wu) assessed multiple outcomes (e.g. all-cause hospitalization and preventable hospitalization) and presented the AUC per outcome. We evaluated and discussed the studies as one model, because study characteristics, methodology and the variables in the final model were identical. However, for the comparison of the predictive accuracy of preventable vs all-cause admissions, we counted these models separately. In other counts we only included the reported AUC for all-cause admission.

• The study of Kurichi et al. presented two different models because of collinearity between the ADL and IADL variable. Predictive performance of the models was equal (AUC = 0.67). The variables included in the final model were equal as well, with the minor difference that the IADL model had an extra variable (i.e. proxy responded). Because of these minor differences, we assessed these two models as one. 

Textual comments

Same as the methods. Results can be pointed out more straightforward. See examples below.

Line 162/168: A flow diagram of the search strategy and selection process is presented in Figure 1, can be removed. Data extracted from the studies can be found in Table 1 and Table 2. Suggest to change it to: The literature searches yielded a total of 16,098 citations (Figure 1.). Tables and figures do not a notification, referring is the standard.

We thank the reviewer for their suggestion and have amended accordingly. (page 7, lines 148, 151-152,

Line 164. Additionally, twenty-three articles were identified through other sources. What are these other sources? This was not mentioned in the methods.

Other sources was reference checking, this is mentioned in the methods section and we have now clarified this in the referred line. (page 7, lines 149-150)

Line 165: In addition to 10 studies included in the previously published systematic reviews, 12 new studies met all inclusion criteria, which makes a total of 22 unique risk prediction models. Rephrase: Full texts were retrieved for 170 studies of which 12 met all inclusion criteria. Additionally, a total of 10 studies were included from the previously published systematic reviews.

We thank the reviewer for their suggestion and we have rephrased accordingly. (page 7, lines 150-152) 

I would suggest to refer to “prediction model” instead of “study” in the results sections. For example line 171: Thirteen studies included participants aged ≥65 172 years[29, 33, 34, 36-40, 44, 46-49], the remaining studies used a higher age as inclusion criterion with. Rephrase to: Thirteen prediction models included……..

We thank the reviewer for their suggestion. However, since some studies developed multiple models with the same data source as we pointed out earlier, we believe studies would be more comprehensible. We did check whether there is consistency in the use of ‘models’ and ‘studies’ and adjusted when necessary.

When describing results, try to hold on to the structure in the methods. The EPV can be described in the “predictive accuracy” section.

The EPV is not a measure to assess performance of the model, but is rather used as a sample size criterion to minimize overfitting of a prediction model. Therefore, it is better gathered under the methodological quality section than the predictive accuracy section.

Avoid using question marks in tables and figures. I would suggest to use NA or a color scheme, for example, high risk= red, low risk= green, unclear= purple.

We thank the reviewer for their suggestion. We however presented the methodological quality assessment according to the suggested tabular presentation of the PROBAST study group, which includes the use of question marks. (see Table 12, Moons et al. [12])

DISCUSSION

I do not understand why the difference between machine learning and logistic regression models is discussed prominently in this article. In order to say whether one technique is superior to the other, you should validate both models in the identical population. In line 317 the authors state the machine learning techniques are not superior and in in line 328 the authors state that a fair comparison is not possible. Please be consequent in conclusions.

We thank the reviewer for their comment. We agree the discussion about the differences between machine learning and logistic regression models is put quite superficially, whereas it is a controversial and complicated topic. We have decided to omit most part of this discussion and put more focus on other results of this review. 

The discussion includes a lot of repetition of the results. Conclusions is solely based on the development of more prediction models. In the introduction the authors describe that the eventual goal is to develop a care management program to avoid these admission. The authors should highlight, how they could use these models to develop such a program. For example; are the variables in these model, standard measurements in a primary care facility?

If the conclusion is only developing more models, the authors should describe how to accomplish this. Which data source to use, which variables should be included, which modeling technique etc.

We thank the reviewer for their suggestion. We have amended the discussion rigorously in light of the comments of the reviewers. We presented suggestions for further improvement of predictive performance and methodological quality of prediction model studies (page 19, lines 295-298; page 20-21, lines 330-352) . Furthermore, we reported clinical implications to the finding that preventable admissions tended to have better predictive performance. (page 21-22, line 354-371)

Textual comments:

Line 293: Twelve risk models were added to the existing evidence. This does not read well. Suggest to remove this sentence.

We thank the reviewer for their comment and have removed the sentence. 

Line 293-295: The recommendation of using nonmedical factors is never mentioned in the manuscript. This conclusion comes a bit out of the blue. I would recommend to make a more general conclusion, on the results that the authors did find (e.g. quality of models, performance of models etc.).

We thank the reviewer for their comment. As mentioned before, we built on the systematic review performed by Wallace et al. One of the recommendations in this review was to consider nonmedical factors for improvement of predictive accuracy. We have made this conclusion more clear in the manuscript. (page 19, lines 277 and further)

 

References:

1. Tsang G, Zhou SM, Xie X. Modeling Large Sparse Data for Feature Selection: Hospital Admission Predictions of the Dementia Patients Using Primary Care Electronic Health Records. IEEE J Transl Eng Health Med. 2021;9:3000113. PubMed PMID: rayyan-128526376.

2. Maust DT, Kim HM, Chiang C, Langa KM, Kales HC. Predicting Risk of Potentially Preventable Hospitalization in Older Adults with Dementia. Journal of the American Geriatrics Society. 2019;67(10):2077-84. PubMed PMID: rayyan-128520734.

3. Moons KG, Altman DG, Reitsma JB, Ioannidis JP, Macaskill P, Steyerberg EW, et al. Transparent Reporting of a multivariable prediction model for Individual Prognosis or Diagnosis (TRIPOD): explanation and elaboration. Annals of internal medicine. 2015;162(1):W1-73. Epub 2015/01/07. doi: 10.7326/m14-0698. PubMed PMID: 25560730.

4. Debray TPA, Damen JAAG, Snell KIE, Ensor J, Hooft L, Reitsma JB, et al. A guide to systematic review and meta-analysis of prediction model performance. BMJ. 2017;356:i6460. doi: 10.1136/bmj.i6460.

5. Group CPM. Cochrane Systematic Review of Prognosis Studies [cited 2022 April 23]. Available from: https://methods.cochrane.org/prognosis/tools.

6. Tian W. An All-Payer View of Hospital Discharge to Postacute Care, 2013: Statistical Brief #205. Healthcare Cost and Utilization Project (HCUP) Statistical Briefs. Rockville (MD): Agency for Healthcare Research and Quality (US); 2006.

7. Kowal P, Dowd J. Definition of an older person. Proposed working definition of an older person in Africa for the MDS Project2001.

8. Andaur Navarro CL, Damen J, Takada T, Nijman SWJ, Dhiman P, Ma J, et al. Protocol for a systematic review on the methodological and reporting quality of prediction model studies using machine learning techniques. BMJ open. 2020;10(11):e038832. Epub 2020/11/13. doi: 10.1136/bmjopen-2020-038832. PubMed PMID: 33177137; PubMed Central PMCID: PMCPMC7661369.

9. Andaur Navarro CL, Damen JAA, Takada T, Nijman SWJ, Dhiman P, Ma J, et al. Risk of bias in studies on prediction models developed using supervised machine learning techniques: systematic review. BMJ. 2021;375:n2281. doi: 10.1136/bmj.n2281.

10. Veyron JH, Friocourt P, Jeanjean O, Luquel L, Bonifas N, Denis F, et al. Home care aides' observations and machine learning algorithms for the prediction of visits to emergency departments by older community-dwelling individuals receiving home care assistance: A proof of concept study. PloS one. 2019;14(8):e0220002. PubMed PMID: rayyan-128526748.

11. de Gelder J, Lucke JA, de Groot B, Fogteloo AJ, Anten S, Mesri K, et al. Predicting adverse health outcomes in older emergency department patients: the APOP study. Neth J Med. 2016;74(8):342-52. Epub 2016/10/21. PubMed PMID: 27762216.

12. Moons KGM, Wolff RF, Riley RD, Whiting PF, Westwood M, Collins GS, et al. PROBAST: A Tool to Assess Risk of Bias and Applicability of Prediction Model Studies: Explanation and Elaboration. Annals of internal medicine. 2019;170(1):W1-w33. Epub 2019/01/01. doi: 10.7326/m18-1377. PubMed PMID: 30596876.

---

## [Decision Letter · Decision Letter 1]

18 Jul 2022

PONE-D-21-38410R1Prediction models for the prediction of unplanned hospital admissions in community-dwelling older adults: a systematic review.PLOS ONE

Dear Dr. Klunder,

Thank you for submitting your manuscript to PLOS ONE. After careful consideration, we feel that it has merit but does not fully meet PLOS ONE’s publication criteria as it currently stands. Therefore, we invite you to submit a revised version of the manuscript that addresses the points raised during the review process.

We look forward to receiving your revised manuscript.

Kind regards,

Dong Keon Yon, MD, FACAAI

Academic Editor

PLOS ONE

Journal Requirements:

Additional Editor Comments:

Please address excellent comments of the reviewers.

Reviewers' comments:

Reviewer's Responses to Questions

**Comments to the Author**

1. If the authors have adequately addressed your comments raised in a previous round of review and you feel that this manuscript is now acceptable for publication, you may indicate that here to bypass the “Comments to the Author” section, enter your conflict of interest statement in the “Confidential to Editor” section, and submit your "Accept" recommendation.

Reviewer #1: All comments have been addressed

Reviewer #3: All comments have been addressed

2. Is the manuscript technically sound, and do the data support the conclusions?

Reviewer #1: Yes

Reviewer #3: Yes

3. Has the statistical analysis been performed appropriately and rigorously? 

Reviewer #1: Yes

Reviewer #3: Yes

4. Have the authors made all data underlying the findings in their manuscript fully available?

Reviewer #1: Yes

Reviewer #3: Yes

5. Is the manuscript presented in an intelligible fashion and written in standard English?

Reviewer #1: Yes

Reviewer #3: (No Response)

6. Review Comments to the Author

Reviewer #1: (No Response)

Reviewer #3: I would like to thank the authors for the revised manuscript. The manuscript has significantly improved and is well structured. The introduction contains all relevant information and emphasizes the relevance of this manuscript. However, I still have minor (profoundly textual comments) that can further improve this paper.

Line 193-196

Very long and hard to follow sentence. Suggest to rephrase.

Line 216

“additionally one study assessed fall with hospitalizations as outcome”.

As mentioned in the exclusion criteria, models developed for specific disease groups were excluded. I would suggest to remove this sentence in order the avoid confusion as the model also looked at ED visit and hospital admission.

Line 276-277

Replace “and” by “an”

line 281-284

suggest to rephrase sentence.

Line 338-339

Instead of “narrowing” I would suggest the term “focusing”.

Discussion

Line 343-344

Suggest to remove the result of AUC>0.8 for fall related hospital admission as suggested earlier. This model performance is namely for fall related hospitalizations.

The start of your discussion includes a lot of comparison with Wallace et al. I would suggest to narrow this part and only highlight the most important difference with an explanation. For exammple, in your results and discussion you mention that the predictive accuracy of the current models has significantly improved compared to the models in Wallace et al. Is there any explanation to this?

You also mention the further implications for future research. Should we indeed develop more models? And what about the nonmedical factors mentioned by Wallace at al. Could the authors maybe elaborate more on this topic in their discussion.

7. PLOS authors have the option to publish the peer review history of their article (what does this mean?). If published, this will include your full peer review and any attached files.

Reviewer #1: No

Reviewer #3: No

---

## [Author Response · Author response to Decision Letter 1]

1 Sep 2022

Reviewer #1: (No Response)

Reviewer #3: I would like to thank the authors for the revised manuscript. The manuscript has significantly improved and is well structured. The introduction contains all relevant information and emphasizes the relevance of this manuscript. However, I still have minor (profoundly textual comments) that can further improve this paper.

Line 193-196 

Very long and hard to follow sentence. Suggest to rephrase.

• We assume the reviewer refers to the following sentence: “For calculations of the median predictive performance we only included the AUC for the outcome all-cause admission, except for the calculation of the median predictive performance per outcome (i.e. preventable and all-cause hospitalization), then AUCs for both outcomes were included.”

We agree this sentence causes confusion and have rephrased it:

“The models developed for a specific type of hospitalization (i.e. preventable hospitalization or fall with hospitalization) (n=3), tended to perform better than the models for all-cause hospitalization (n=17), with a median AUC of 0.78 (range 0.74-0.78) versus 0.69 (range 0.61 - 0.76), respectively. The two models that assessed AUCs for both outcomes (i.e. Tarekegn et al. and Wu et al.[37, 39]) were included in calculations of both medians with its corresponding AUC and were thus counted twice.”

Line 216

“additionally one study assessed fall with hospitalizations as outcome”.

As mentioned in the exclusion criteria, models developed for specific disease groups were excluded. I would suggest to remove this sentence in order the avoid confusion as the model also looked at ED visit and hospital admission.

• Thank you for your comment. We have removed any reference to the fall-related hospital admissions outcome of this model from the rest of the manuscript, as it is indeed an exclusion criterion mentioned in the methods section. 

Line 276-277

Replace “and” by “an”

• Thank you for noticing the typo.

line 281-284

suggest to rephrase sentence.

• We have rephrased the sentence accordingly:

“Concern for applicability was high in three studies, because the study population or study outcome did not fully match the review question: one study only included older adults with a sensory impairment[29], one study excluded older adults with a hospital admission <6 months prior to the index date[45], and one study evaluated preventable hospital admissions as only outcome[46].”

Line 338-339

Instead of “narrowing” I would suggest the term “focusing”.

• Thank you, we have followed your suggestion.

Discussion

Line 343-344

Suggest to remove the result of AUC>0.8 for fall related hospital admission as suggested earlier. This model performance is namely for fall related hospitalizations.

• We have removed this sentence in the discussion.

The start of your discussion includes a lot of comparison with Wallace et al. I would suggest to narrow this part and only highlight the most important difference with an explanation. For example, in your results and discussion you mention that the predictive accuracy of the current models has significantly improved compared to the models in Wallace et al. Is there any explanation to this?

• Thank you for your suggestion. We have narrowed this paragraph and included a possible explanation to the increased predictive accuracy in the newer models. 

“The new models had higher predictive accuracy than the older models. This might be explained by the fact that new models had larger sample sizes of the development cohort and also higher EPVs. Both are recommended by the TRIPOD, published in 2015 [16], to improve predictive accuracy and methodological quality. Moreover, the new models used administrative or clinical record data more often for the development of their model. Consistent with Wallace et al., we found that models developed using administrative or clinical record data had higher predictive accuracy than those developed using self-report data. Of the 10 new prediction models, eight used administrative data for development of their model.”

You also mention the further implications for future research. Should we indeed develop more models? And what about the nonmedical factors mentioned by Wallace at al. Could the authors maybe elaborate more on this topic in their discussion.

• Thank you. We have followed your suggestions and have rephrased the implications for future research paragraph. We put more emphasis on external validation and updating prediction models instead of developing new prediction models. Regarding updating of the models, we suggested the addition of nonmedical factors may contribute to an improved accuracy of the model. (Manuscript, pages 19-20, lines 320-344)

---

## [Decision Letter · Decision Letter 2]

8 Sep 2022

PONE-D-21-38410R2Prediction models for the prediction of unplanned hospital admissions in community-dwelling older adults: a systematic review.PLOS ONE

Dear Dr. Klunder,

Thank you for submitting your manuscript to PLOS ONE. After careful consideration, we feel that it has merit but does not fully meet PLOS ONE’s publication criteria as it currently stands. Therefore, we invite you to submit a revised version of the manuscript that addresses the points raised during the review process.

We look forward to receiving your revised manuscript.

Kind regards,

Dong Keon Yon, MD, FACAAI

Academic Editor

PLOS ONE

Journal Requirements:

Additional Editor Comments:

This is an excellent paper. Finally, please replace reference number 19 (PRISMA guideline 2009) with the following recent paper (PRISMA guideline 2020).

DOI: https://doi.org/10.54724/lc.2022.e9

Congratulations!

Reviewers' comments:

Reviewer's Responses to Questions

**Comments to the Author**

1. If the authors have adequately addressed your comments raised in a previous round of review and you feel that this manuscript is now acceptable for publication, you may indicate that here to bypass the “Comments to the Author” section, enter your conflict of interest statement in the “Confidential to Editor” section, and submit your "Accept" recommendation.

Reviewer #1: All comments have been addressed

Reviewer #3: All comments have been addressed

2. Is the manuscript technically sound, and do the data support the conclusions?

Reviewer #1: Yes

Reviewer #3: Yes

3. Has the statistical analysis been performed appropriately and rigorously? 

Reviewer #1: Yes

Reviewer #3: Yes

4. Have the authors made all data underlying the findings in their manuscript fully available?

Reviewer #1: Yes

Reviewer #3: Yes

5. Is the manuscript presented in an intelligible fashion and written in standard English?

Reviewer #1: Yes

Reviewer #3: Yes

6. Review Comments to the Author

Reviewer #1: (No Response)

Reviewer #3: I would like to thank the authors for their revised mansucript. All comments are adressed. Readability and structure have improved, making the manuscript ready for publication.

7. PLOS authors have the option to publish the peer review history of their article (what does this mean?). If published, this will include your full peer review and any attached files.

Reviewer #1: No

Reviewer #3: No

---

## [Author Response · Author response to Decision Letter 2]

8 Sep 2022

Thank you for the quick and thorough review and the opportunity to publish in PLOS One. We have replaced reference number 19 with the suggested recent paper. I have uploaded the revised manuscript file. I did not include a Manuscript with Track Changes file, since MS Word does not track any changes in the reference list. If you need any additional information/files, please let me know.

---

## [Editor Report · Decision Letter 3]

11 Sep 2022

Prediction models for the prediction of unplanned hospital admissions in community-dwelling older adults: a systematic review.

PONE-D-21-38410R3

Dear Dr. Klunder,

We’re pleased to inform you that your manuscript has been judged scientifically suitable for publication and will be formally accepted for publication once it meets all outstanding technical requirements.

Kind regards,

Dong Keon Yon, MD, FACAAI

Academic Editor

PLOS ONE

Additional Editor Comments (optional):

This is an excellent and mesmerzing paper.
---

## [Editor Report · Acceptance letter]

15 Sep 2022

PONE-D-21-38410R3 

Prediction models for the prediction of unplanned hospital admissions in community-dwelling older adults: a systematic review. 

Dear Dr. Klunder:

I'm pleased to inform you that your manuscript has been deemed suitable for publication in PLOS ONE. Congratulations! Your manuscript is now with our production department. 

Kind regards, 

on behalf of

Dr. Dong Keon Yon 

Academic Editor

PLOS ONE